# Index microvascular resistance (IMR) in heart transplant patients (IMR-HT study): Study protocol

Ainhoa Pérez-Guerrero[1,2,3☯], Jean Paul Vilchez-Tschischke[4], Luis Almenar Bonet[4,5], Jose Luis Diez Gil[4,5], Teresa Blasco Peiró[1], Salvatore Brugaletta[6], Josep Gomez-Lara[7], José González Costello[5,7], Paula Antuña[8], Vanesa Alonso Fernández[8], Fernando Sarnago Cebada[9], María Dolores García-Cosio[5,9], Francisco Hidalgo Lesmes[10], Amador López Granados[10], Ramón López-Palop[11], Iris Paula Garrido[11], Rosa María Cardenal Piris[12], Diego Rangel Sousa[12], Georgina Fuertes Ferre[1,3☯]*

1 Cardiology, Miguel Servet University Hospital, Zaragoza, Spain, 2 Clínico Lozano Blesa University Hospital, Zaragoza, Spain, 3 Aragon Health Research Institute (IIS Aragon), Zaragoza, Spain, 4 La Fe University Hospital, Valencia, Spain, 5 Center for cardiovascular biomedical research (CIBER-CV), Madrid, Spain, 6 Hospital Clínic, Cardiovascular Clinic Institute, Institut d'Investigacions Biomèdiques August Pi i Sunyer (IDIBAPS), University of Barcelona, Barcelona, Spain, 7 Bellvitge University Hospital, IDIBELL, University of Barcelona. L'Hospitalet de Llobregat, Barcelona, Spain, 8 Asturias Central University Hospital, Oviedo, Spain, 9 12 Octubre University Hospital, Imas12, Madrid, Spain, 10 Reina Sofía de Córdoba University Hospital, Córdoba, Spain, 11 Virgen de la Arrixaca University Hospital, Murcia, Spain, 12 Virgen del Rocío University Hospital, Sevilla, Spain

☯ These authors are equal contributors to this work.
* georginaff@hotmail.com

## Abstract

### Background

Acute allograft rejection (AAR) is an important cause of morbi mortality in heart transplant (HT) patients, particularly during the first year. Endomyocardial biopsy (EMB) is the "gold standard" to guide post- heart transplantation treatment. However, it is associated with complications that can be potentially serious. The index of microvascular resistance (IMR) is a specific physiological parameter used to assess microvascular function. Invasive coronary assessment has been shown to be both feasible and safe. Detection of coronary microvascular dysfunction (MCD) by IMR may help to identify high risk HT patients. In fact, an increased IMR measured early after HT has been associated with AAR, higher all-cause mortality and adverse cardiac events. A high IMR value early after HT may identify patients at higher risk who require increased surveillance or adjustments in immunosuppressive therapy. Conversely, a low IMR value may support reducing the number of EMBs. Our aim is to evaluate IMR in heart transplant patients within the first year. Changes in management after knowing IMR values and prognostic implications of IMR in a long term follow up will also be assessed.

### Study design

The IMR-HT study (NCT 06656065) is a multicenter, prospective study that will include post-HT consecutive stable patients undergoing coronary physiological assessment in the

**Data availability statement:** Deidentified research data will be made publicly available when the study is completed and published.

**Funding:** The author(s) received no specific funding for this work.

**Competing interests:** The authors have declared that no competing interests exist.

**Abbreviations: AAR**: acute allograft rejection; **ACR**: acute cellular rejection; **CAV**, cardiac allograft vasculopathy; **CFR**, coronary flow reserve; **EMB**, endomyocardial biopsy; **FFR**, fractional flow reserve; **HT**, heart transplant; **IMR**, index of microcirculatory resistance; **IST**, immunosuppressive therapy; **MVD**, microvascular dysfunction; **Pa**, aortic pressure; **Pah**, aortic pressure at hyperemia; **Pd**, distal coronary pressure; **Pdh**, distal coronary pressure at hyperemia; **RFR**, resting full-cycle-ratio; **Tmnh**, mean transit time at hyperemia; **Tmnr**, mean transit time at rest

first three months and one year. Cardiac adverse events will be evaluated at one year for up to five years. A clinical management algorithm is proposed: after knowing IMR values the physician will be able to reduce the number of biopsies established in each center protocol or modify immunosuppression therapy.

## Conclusions

IMR values may vary within the first year after heart transplant. IMR assessment will be useful to identify high risk heart transplant patients, leading to possible changes in management and prognosis.

## Introduction

Acute allograft rejection (AAR) is an important cause of morbi mortality after heart transplant (HT), particularly within the first year[1–3]. Advances in immunosuppression, donor heart evaluation, surgical techniques, and post-transplantation care have led to a gradual reduction in AAR and improved survival after HT over time. Endomyocardial biopsy (EMB) is the gold standard method to guide post-HT treatment, as it represents the best tool to identify rejection in orthotopic HT [4–5]. However, it is usually repeated up to 5 times during the first year - with some variations depending on each center protocol - and it is potentially associated with serious complications.

Several studies have presented the association between AAR, micro-vasculopathy and cardiac allograft epicardial vasculopathy (CAV) [6–9]. Index of microcirculatory resistance (IMR) measured early after heart transplantation has been significantly associated with the risk of acute cellular rejection (ACR), and patients with IMR ≥ 15 have higher risk of AAR for 2 years follow-up [10].

Our aim is to evaluate IMR in heart transplant patients within the first year. Changes in management after knowing IMR values and prognostic implications of IMR in a long term follow up will also be assessed.

## Methods

### Rationale and design

The IMR-HT study is a multicenter, prospective study aimed to assess IMR values within the first year after heart transplant.

Changes in HT patient management (number of EMBs, immunosuppressive therapy modifications) after knowing IMR values will also be assessed, as well as the prognostic implications of IMR baseline and annual values.

It will enroll consecutive eligible patients who undergo HT in each participation center.

### Hypothesis

There are few studies that evaluate IMR after a heart transplant. High IMR baseline value will identify patients at higher risk and will lead to modifications in the number of biopsies and/or immunosuppressive therapy within one year. Patients with an increase in IMR over the first year will have a higher rate of cardiac events.

### Patient selection

Patients undergoing HT will be screened for enrollment. To maximize patient inclusion, we apply broad inclusion criteria and strict exclusion criteria, as specified in **Table 1**. Eligible

**Table 1. Eligibility criteria.**

**Inclusion criteria**

Patients ≥ 18 years old who undergo heart transplantation in participant's centers.

Informed consent received and signed for study enrollment.

**Exclusion criteria**

Patients with hemodynamic instability after HT, including cardiogenic shock or severe coagulopathy.

Patients with acute cellular rejection before intracoronary physiological assessment.

Patients with bronchial asthma or bronchopathy with a positive bronchodilation test, which contraindicate the use of adenosine.

Patients with epicardial coronary lesions with a resting physiological index ≤ 0.89 or in hyperemia ≤ 0.80.

Patients unlikely to cooperate in the study or with inability or unwillingness to give informed consent.

patients will be informed about the study and will have to provide written informed consent prior to being included. The recruitment begins on May 23rd, 2023, the expected recruitment period is two years. The study completion date is estimated in 2030. This study adhered to the principles outlined in the Declaration of Helsinki. The approval was granted by the Ethics Committee for Investigation of Aragon (CEICA). The study has been registered at Clinical-Trials.gov (NCT 06656065).

The schedule of enrollment, interventions and assessments and the timeline of the study are detailed in **Fig 1**.

| | STUDY PERIOD | | | | | | | | |
|---|---|---|---|---|---|---|---|---|---|
| | Enrolment | Allocation | Post-allocation | | | | | Close-out | |
| **TIMEPOINT**\*\* | **-$t_1$** | **0** | **$t_1$** | **$t_2$** | **$t_3$** | **$t_4$** | **etc.** | **$t_x$** | |
| **ENROLMENT:** | | | | | | | | | |
| **Eligibility screen** | X | - | | | | | | | |
| **Informed consent** | X | - | | | | | | | |
| *[List other procedures]* | X | - | | | | | | | |
| **Allocation** | N/A | | | | | | | | |
| **INTERVENTIONS:** | | | | | | | | | |
| *[Intervention A]* | N/A | | | | | | | | |
| *[Intervention B]* | N/A | | | | | | | | |
| *[List other study groups]* | N/A | | | | | | | | |
| **ASSESSMENTS:** | | | | | | | | | |
| *[List baseline variables]* | X | - | | | | | | | |
| *[List outcome variables]* | X | - | | | | | | | |
| *[List other data variables]* | X | - | | | | | | | |

\*Recommended content can be displayed using various schematic formats. See SPIRIT 2013 Explanation and Elaboration for examples from protocols.

\*\*List specific timepoints in this row.

**Fig 1. Schedule of enrolment, interventions, and assessments.**

Clinical conditions, laboratory findings and clinical events will be assessed at one month and one year. Follow up will be extended for up to five years.

**Invasive physiological evaluation of the coronary microcirculation.** Patients included will undergo a coronary angiography between the first and third month after HT. This coronary angiography is part of the follow-up protocol in most centers also including EMB (Fig 2).

Assessment of IMR, coronary flow reserve (CFR) and fractional flow reserve (FFR) will be performed using the standard technique [11–13]. The left anterior descending coronary artery will be evaluated in all patients. Circumflex or right coronary artery could be additionally evaluated at operator's discretion. An intracoronary pressure and temperature sensor-tipped guidewire (Pressure Wire ™ X guide- wire 0.014', Abbott, IL, USA) will be used to perform the measurements. The tip pressure sensor will be advanced into the mid-to-distal portion of the evaluated vessel (50 to 60 mm of the ostium of selected coronary artery). Baseline aortic pressure (Pa) and distal intracoronary pressure (Pd) will be obtained to calculate the resting index Pd/Pa. To measure the mean transit time (Tmn) under basal conditions, intracoronary administration of 3 mL of room-temperature saline will be manually injected three times in succession (3 mL/s). Then maximal hyperemia will be induced using adenosine iv (140 to 180 mg/kg/min) and three additional intracoronary room temperature saline boluses of 3 ml will be administered to determine the mean transit time at hyperemia (Tmnh). Deviations > 10% in some of the individual Tmn values will force their repetition. Both at rest and in hyperemia, the mean of the three individual determinations will be used for the calculations. Finally, fractional flow reserve (FFR), coronary flow reserve (CFR) and IMR will be calculated using the software Coroventis Coroflow (*Abbott, Coroventis Uppsala, Sweden)*. Physiological indexes are listed in **Table 2**.

- Fractional flow reserve (FFR) is defined as the ratio of maximal coronary blood flow in a diseased artery to the maximal coronary blood flow in the same artery without stenosis. FFR serves as a surrogate marker of inducible myocardial ischemia caused by epicardial coronary stenosis.

- Coronary flow reserve (CFR) is the ratio of hyperemic to baseline coronary blood flow and acts as a marker of the integrity of both epicardial and microvascular coronary circulation. Therefore, CFR reflects microvascular status in the absence of significant epicardial disease.

- Index of Microcirculatory Resistance (IMR) represents the minimum achievable resistance in the coronary microcirculation and is a more specific marker of microvascular function.

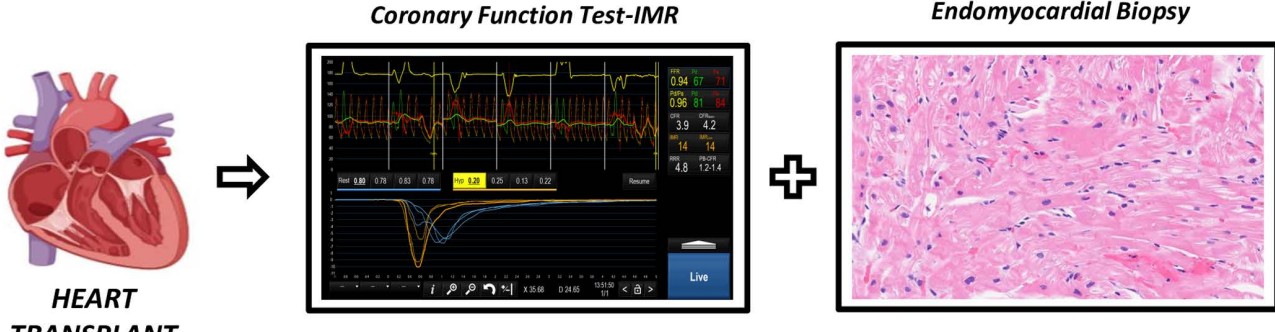

**Fig 2. Coronary function test and endomyocardial biopsy between the first and third month after heart transplant.**

**Table 2. Calculation FFR, IMR and CFR.**

| FFR = Pdh/ Pah |
| --- |
| FFR: fractional flow reserve<br>Pdh: distal coronary pressure at hyperemia<br>Pah: aortic pressure at hyperemia |
| IMR = Pd × Tmnh |
| IMR: index of microcirculatory resistance<br>Pd: distal coronary pressure<br>Tmnh: mean transit time at hyperemia |
| CFR = CFR = 1/(Tmnh/Tmnr) |
| CFR: Coronary flow reserve<br>Tmnh: mean transit time at hyperemia<br>Tmnr: mean transit time at rest |

IMR is defined as the distal coronary pressure divided by the inverse of the hyperemic mean transit time and is expressed in units of mmHg.s.

The physiological assessment of IMR, CFR, and FFR will be repeated one year after HT.

**Biopsies and immunosuppressive treatment.** Based on previously published clinical data on IMR in heart transplant patients, a post-HT management algorithm is proposed:

- IMR < 15: The frequency of biopsies could be reduced or maintained as per protocol. No changes to immunosuppressive therapy would be required.

- IMR ≥ 15: Biopsies would be performed at the standard frequency according to protocol. Immunosuppressive therapy could be intensified or maintained the same.

The number of biopsies performed by each center, as well as changes in immunosuppressive therapy after knowing IMR baseline values will be assessed. Of note, given the observational characteristics of the study, clinical management decisions will be made at the discretion of the treating physician, considering the patient's clinical condition and other complementary tests.

Both groups (IMR < 15 vs IMR ≥ 15) will be compared in terms of cardiac events occurrence.

The prognostic implications of IMR variation (if that is the case) between the first three months and one year will also be analyzed.

The results of this observational trial will help in the conduct of a randomized trial in which an EMB will be spared based on the IMR value.

The proposed study algorithm is shown in **Fig 3**. Participating centers are listed in **Supplementary Table 1**.

**Data collection.** Sociodemographic, clinical, laboratory and follow-up data of each patient will be included in a database specifically designed for the study. All variables will be included in the online data collection platform Redcap (Research Electronic Data Capture). Each patient will be assigned a number; their identity will not be revealed in any case. All shared information will be anonymized.

**Study endpoints.** The **main objective** of the study is to evaluate IMR values in the first three months and one year after heart transplant. Secondary endpoints are detailed in **Table 3**. End-points will be evaluated at 1 year and annually thereafter for up to 5 years. Definition of events is detailed in **Table 4**.

**Statistical analysis and sample size considerations.** The characteristics of the study population will be summarized using standard descriptive statistics. Continuous variables will

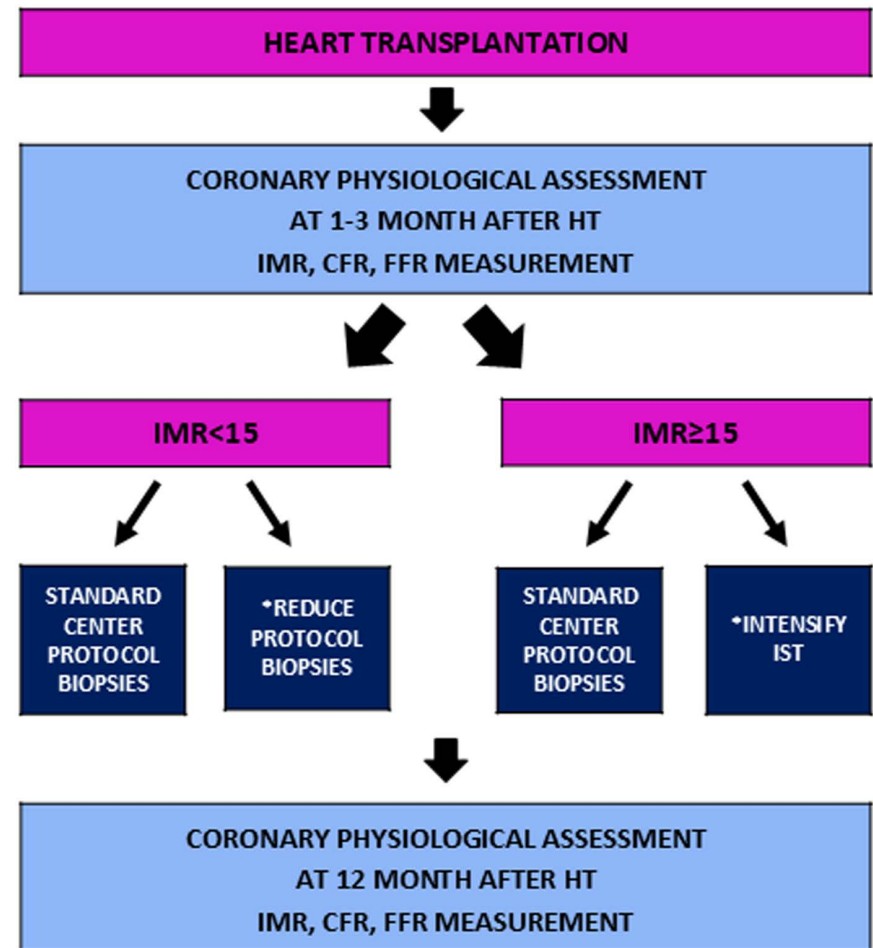

**Fig 3. Study proposed algorithm.**

be expressed as mean ± standard deviation (SD) or median [interquartile range (IQR)], as appropriate. Categorical variables will be presented as absolute numbers and percentages. For the comparison of means, the Student's t-test for independent samples or the non-parametric Mann-Whitney U test will be used for dichotomous qualitative variables. For non-dichotomous qualitative variables, the ANOVA test or the non-parametric Kruskal-Wallis test will be applied.

**Table 3. Secondary endpoints.**

| Secondary endpoints |
| --- |
| Combined endpoint of CV mortality, cardiac allograft vasculopathy, acute cellular rejection, hospitalization for heart failure, re-transplantation at 1-year and IMR. |
| Cardiovascular mortality and IMR. |
| Cardiac allograft vasculopathy and IMR. |
| Acute cellular rejection and IMR. |
| Number of biopsies performed in each center within the first year after knowing initial IMR results. |
| Modifications in immunosuppressive therapy after knowing initial IMR results. |

**Table 4. Definition of events.**

| Acute cellular rejection (ACR) |
| --- |
| Grade ≥ 2R by the 2010 ISHLT grading system. Finding in the endomyocardial biopsy of two or more foci of infiltrates associated with myocyte damage. Diffuse infiltration with multifocal myocyte damage with/without edema, hemorrhage or vasculitis. |
| Cardiovascular mortality |
| Caused by cardiovascular disease or unknown death. |
| Cardiac allograft vasculopathy (CAV) |
| Accelerated fibroproliferative process characterized by a diffuse, concentric and longitudinal thickening of the intima of the vascular tree of the graft, affecting everything from the great epicardial arteries to the coronary microvasculature |
| Combined safety endpoint |
| Acute cellular rejection, heart failure, re-transplantation and cardiovascular mortality. |

For bivariate analysis of qualitative variables, the Chi-square test or Fisher's exact test will be used, as appropriate. Event comparisons between groups will be performed using Kaplan-Meier curves, and a Cox regression analysis will be conducted to adjust for confounding factors and identify independent predictors of clinical events. A two-tailed p-value < 0.05 will be considered statistically significant. All statistical analyses will be performed using SPSS software.

Given the exploratory nature of the study, no formal power calculation is required. In 2023, approximately 325 heart transplants (HT) were performed in Spain. With a median of 10–15 HTs per center and considering the participation of eight centers that routinely perform invasive coronary physiology, data will be analyzed once 100 patients have completed their first-year follow-up [3]

## Discussion

Heart transplant is considered the treatment of choice in patients with advanced heart failure refractory to medical treatment or devices [1–2].

AAR plays an important role in determining prognosis: up to 20% of HT patients experience at least one episode of ACR in the first-year post-transplant [1]. The immune response is classified into ACR when it is mediated by T lymphocytes and humoral rejection when the main mechanism involves B lymphocytes and antibody production [4,5,14]. Advances in immunosuppressive therapy (IST), donor heart evaluation, surgical techniques and post-HT care have led to a reduction in ACR, improving survival over time [4]. Post-transplant IST includes three basic components: a calcineurin inhibitor (currently preferred Tacrolimus), an antiproliferative agent (mycophenolate mofetil), and steroids. On the other hand, proliferation signal inhibitor (mTOR) drugs (everolimus and sirolimus) are primarily used for CAV [15]. CAV is the main cause of mortality after the first year of transplantation. It is characterized by diffuse intimal thickening affecting both coronary epicardial and microcirculation [4,15].

The vast majority of ACR occur asymptomatically, presenting normal ventricular function, and thus being detected through the routine EMB surveillance protocol [4]. The graduation of the RAC is detailed in the 2005 review of the ISHLT [14]. Due to intra- and inter-observer variabilities in determining different degrees of slight-moderate rejection, an update was published in the 2010 document. In the International Society of Heart and Lung Transplantation (ISHLT) guidelines, EMB was made a IIaC recommendation for the detection of rejection [16].

EMB is repeatedly performed during the first year after HT. It is often described as uncomfortable by patients, and is associated with complications that can be potentially serious, such is the case of cardiac perforation.

Moreover, EMB diagnostic yield is wide, with a high variability between observers and a non-negligible rate of false positives and negatives [17].

To avoid the inconveniences of EMB, non-invasive techniques such as cardiac magnetic resonance have been studied for detecting rejection. In recent years, advanced methods like gene expression profiling and plasma donor-derived cell-free DNA have also been introduced. However, their results have been highly variable, and these techniques have not been implemented in routine clinical practice in most of our centers [18,19]. For these reasons, none of these techniques has been able to replace EMB. IMR is a quantitative and specific index of coronary microcirculation that requires the use of a coronary guide and hyperemia to be analyzed. Both adenosine and the use of a coronary wire could cause adverse symptoms [20]. However, the rate of potential serious adverse events is equivalent to that of routine diagnostic coronary angiography (<1%) [21]. Particularly in heart transplant patients, Duran SR et al. showed no complications when using adenosine at a rate of 140 mcg/kg/min in 16 pediatric patients undergoing MR stress perfusion cardiac magnetic resonance [22]. There are no studies comparing IMR with other non-invasive tools for the detection of AAR. However, rather than a substitute, IMR may be a complementary technique for rejection surveillance in HT patients.

Some concerns may arise regarding the performance of an invasive microvascular function test, as it requires the use of a coronary guide and, consequently, heparin administration along with EMB extraction in the same procedure. However, this approach has been successfully implemented in other trials aimed at identifying CAV through intracoronary imaging techniques in HT patients, without any significant complications [23].

An increased IMR in the graft has been associated with higher all-cause mortality and adverse cardiac events regardless of epicardial vasculopathy. Several IMR cut-off (from 15 to > 20) have been associated with ACR within the first year after HT. Patients in whom IMR decreases or does not change one year after HT have a higher event-free rate than those patients in whom the IMR increases [8–10,24,25]. As far as we know, no study has evaluated IMR impact on post-HT management. A high IMR value early after HT may detect a higher risk patient that needs an increased surveillance or changes in immunosuppressive therapy (an earlier administration of mTor-inhibiting drugs or prescribing calcium antagonists, which are known to improve microvascular function). On the other hand, a low IMR value may lead to decrease the number of EMB.

Our aim will be to assess IMR values in heart transplant patients within one year and evaluate changes in management after knowing of IMR values. We believe it is important to move forward in AAR surveillance and reduce the number of endomyocardial biopsies. In addition to assessing their diagnostic capabilities, IMR should also be assessed based on clinical outcomes. Therefore, we are convinced the results of this trial will be very important for our HT patient population.

## Conclusions

The IMR is a quantitative physiological parameter to evaluate coronary microcirculation. High IMR values have been associated with acute cellular rejection in heart transplant patients. The assessment of IMR may be useful for identifying high-risk heart transplant patients, leading to changes in management and prognosis.

## Supporting information

**S1 File. SPIRIT checklist**
(DOCX)

**S2 Table. Site centers and investigators.**
(DOCX)

## Acknowledgments

We would like to thank Dr. Mailen Guerrero, from the Anatomical Pathology Department for her contribution to this project.

## Author contributions

**Conceptualization:** Jean Paul Vilchez-Tschischke, Teresa Blasco Peiró, Georgina Fuertes Ferre.

**Data curation:** Ainhoa Pérez-Guerrero, Jean Paul Vilchez-Tschischke, Salvatore Brugaletta.

**Formal analysis:** Ainhoa Pérez-Guerrero, Salvatore Brugaletta.

**Funding acquisition:** Teresa Blasco Peiró, Georgina Fuertes Ferre.

**Investigation:** Ainhoa Pérez-Guerrero, Teresa Blasco Peiró, Salvatore Brugaletta, Paula Antuña, Georgina Fuertes Ferre.

**Methodology:** Ainhoa Pérez-Guerrero, Salvatore Brugaletta, Georgina Fuertes Ferre.

**Project administration:** Teresa Blasco Peiró, Georgina Fuertes Ferre.

**Resources:** Teresa Blasco Peiró, Georgina Fuertes Ferre.

**Supervision:** Ainhoa Pérez-Guerrero, Jean Paul Vilchez-Tschischke, Luis Almenar Bonet, Jose Luis Diez Gil, Salvatore Brugaletta, Josep Gomez-Lara, Vanesa Alonso Fernández, María Dolores García-Cosio, Amador López Granados, Ramón López-Palop, Iris Paula Garrido, Rosa María Cardenal Piris, Diego Rangel Sousa, Georgina Fuertes Ferre.

**Validation:** Ainhoa Pérez-Guerrero, Jean Paul Vilchez-Tschischke, Luis Almenar Bonet, Jose Luis Diez Gil, Salvatore Brugaletta, Josep Gomez-Lara, Paula Antuña, Vanesa Alonso Fernández, Fernando Sarnago Cebada, María Dolores García-Cosio, Francisco Hidalgo Lesmes, Amador López Granados, Ramón López-Palop, Iris Paula Garrido, Georgina Fuertes Ferre.

**Visualization:** Ainhoa Pérez-Guerrero, Jean Paul Vilchez-Tschischke, Jose Luis Diez Gil, Josep Gomez-Lara, Paula Antuña, Vanesa Alonso Fernández, Fernando Sarnago Cebada, María Dolores García-Cosio, Francisco Hidalgo Lesmes, Amador López Granados, Ramón López-Palop, Iris Paula Garrido, Georgina Fuertes Ferre.

**Writing – original draft:** Ainhoa Pérez-Guerrero, José González Costello, Georgina Fuertes Ferre.

**Writing – review & editing:** Ainhoa Pérez-Guerrero, Jean Paul Vilchez-Tschischke, Luis Almenar Bonet, Teresa Blasco Peiró, Salvatore Brugaletta, Josep Gomez-Lara, José González Costello, María Dolores García-Cosio, Ramón López-Palop, Georgina Fuertes Ferre.

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
