## [Decision Letter · Decision Letter 0]

14 Jan 2025

PONE-D-24-52635Index microvascular resistance (IMR)-guided management of heart transplantation: Study protocol

PLOS ONE

Dear Dr. Fuertes Ferre,

Thank you for submitting your manuscript to PLOS ONE. After careful consideration, we feel that it has merit but does not fully meet PLOS ONE’s publication criteria as it currently stands. Therefore, we invite you to submit a revised version of the manuscript that addresses the points raised during the review process.

While the study rationale is well-presented, several issues need clarification. The risks associated with using saline injections and adenosine for IMR assessment are not discussed, and there is no comparison of IMR with other noninvasive diagnostic tools. The hypotheses are framed using non-refutable language, and clearer, more testable formulations are suggested to strengthen the study’s objectives.

The methodology for assigning patients to comparison groups lacks clarity, particularly regarding the configuration of groups based on IMR values and changes in EMB or immunosuppressive therapy. It is recommended to compare outcomes where IMR influenced treatment decisions or to analyze data against historical cases from the same centers. Additionally, discrepancies between the main file and the protocol file, including missing sections, inconsistent references, and grammatical errors, need to be resolved. The definition of IMR based on mean transit time should be consistently presented across documents, and the study’s data sharing plan should specify where data will be available after publication.

We look forward to receiving your revised manuscript.

Kind regards,

Giuseppe Andò, M.D., Ph.D.

Academic Editor

PLOS ONE

5. Please include a caption for figure 2 and 3.

Additional Editor Comments (if provided):

Reviewers' comments:

Reviewer's Responses to Questions

**Comments to the Author**

1. Does the manuscript provide a valid rationale for the proposed study, with clearly identified and justified research questions?

Reviewer #1: Yes

Reviewer #2: Partly

2. Is the protocol technically sound and planned in a manner that will lead to a meaningful outcome and allow testing the stated hypotheses?

Reviewer #1: Yes

Reviewer #2: Partly

3. Is the methodology feasible and described in sufficient detail to allow the work to be replicable?

Reviewer #1: Yes

Reviewer #2: Yes

4. Have the authors described where all data underlying the findings will be made available when the study is complete?

Reviewer #1: Yes

Reviewer #2: No

5. Is the manuscript presented in an intelligible fashion and written in standard English?

Reviewer #1: Yes

Reviewer #2: Yes

6. Review Comments to the Author

You may also provide optional suggestions and comments to authors that they might find helpful in planning their study.

Reviewer #1: Interesting paper

Some issues shiuld be added

Abstract; the correlation between number of byiopsies and IMR is not clear

Methods: adenosine is quite used, do authors have experience about regadenason?

Discussion: personally I find this proposal very interesting. My concern is:are authors sure that performing 2 microcircle evaluation per year is safer/ more cost effective than performin EBM? and bow many EBMs do they think will be spared?

Methods: sample size calculation is needed

Reviewer #2: Fuertes Ferré et al propose to carry out a multi-center prospective observational study on the influence of IMR assessment on the treatment and outcomes of heart transplant patients. Their functional hypothesis is that IMR assessment would reduce the number of endomyocardial biopsies (EMB) which are usually performed to evaluate the risk of acute rejection (AAR). The proposal relies mostly on the work of Lee et al (2021) who studied 154 patients undergoing heart transplantation and established that an index of IMR ≥ 15 does part a population with high risk of AAR (~34%) vs a population with low AAR risk (~4%).

The rationale for the study is well presented, however, the criteria for application of the IMR (>15<) and the assignment of patients to different groups to perform statistical comparisons are not fully clear, as detailed below.

Major concerns

Treatments: It is clearly stated that all HT patients must run through invasive coronary assessment as part of their follow up treatment, which currently includes repetitive EMB. The risks of performing repetitive EMB are discussed, emphasizing the notion that reducing EMB would be a favorable outcome for patients. However, a sound analysis pondering the risks (if any) of performing repetitive room temperature saline injections and adenosine infusion to cause maximal hyperemia in HT patients are not discussed, further, a pondered discussion of pro/con features of IMR versus other noninvasive diagnostic tools is missing.

Hypothesis: One major concern is the “softness” of the propositions used int the hypotheses. Propositions like “could reduce” or “could improve” are hardly refutable. I strongly advice to use a more direct, clearer, easy to assess hypothesis, like “The availability of IMR data in heart transplant patients will reduce the number of ECM performed during the first year … etc. Likewise, “IMR data will improve early treatment in patients with high IMR scores... etc. When the observational study is completed, these hypotheses may or may not be held right.

Experimental Groups for comparisons: Surgical/treating teams in the different Centers will be advised and informed on the protocol, enrolling 100 patients for the study. It is understandable that the physician team at every center should decide on the actual treatment applied to each patient, as stated: “Given the observational characteristics of the study, decisions regarding clinical patient management will depend on physician discretion considering patient clinical condition and other complementary tests.”

This, however, leads to a major problem with the proposal. The “Protocol”, states that two (or perhaps more) groups will be compared. The statistical analysis correctly described different statistical tools to be used according to the characteristics of the variable (continuous or discrete) and their distribution. What is not fully clear is how the composition of groups to be compared will be made (either a priori or a posteriori).

According to Figure 3 of the main submission (same figure in “Protocol”), it is assumed that the main groups to be compared are patients with IMR <15 vs patients with IMR >15. Yet, each group is subdivided into “standard” and “reduced” (for EMB) or “standard” or “intensified” (for immunosuppressive therapy, IST). Since the idea is to assess the impact of knowing IRM scores to proceed on whether performing ECB, how will the groups be configured? And which group will be compared to which one? Will it be four groups in total? (i.e. IMR<15 WITH reduction in EMB; IMR<15 WITHOUT reduction in EMB; IMR>15 WITH improved/early IST; and IMR>15 WITHOUT improved/early IST)?

- Do the authors consider comparing the cases in which knowing the IRM score (above or below 15) changed the criteria for the currently/previously applied treatment?

In short, how will the patients be assigned to different groups in the posterior analysis? And how do we know the number of patients where the criteria for ECB were replaced because of an IRM score >15< ?

Additionally, perhaps the authors consider comparing data of the enrolled patients (i.e. patients with IMR assessment from this study) with previous data of the same Centers (i.e. without IRM assessment) and then analyzing the clinical outcomes.

Minor/Specifics

There are discrepancies between different files composing this submission. The main file is much clearer and precise than the “Protocol” file. Although most of the text is repeated in both files, the main file does not contain important sections, such as the Hypothesis, which is found in “Protocol” and the reviewer must jump from file to file to assess the proposal.

The “Protocol” file was less edited than the main file, opening concerns about Q5 on the review form (i.e. written English standards and intelligible fashion). For instance, some abbreviations are missing, and some references are presented with the same number (i.e. two works are cited as ref #10, etc.).; This file contains several grammatical mistakes, and it must be corrected, hopefully making it to coincide with the main file (references, abbreviations, definitions).

The authors claim that data will be available after publication of the study, but they don’t state where these data will be available, as required by the journal.

In protocols, IMR is defined as: “It is calculated as the ratio of distal coronary pressure to coronary flow at hyperemia and presented in units (mmHg.s)” Although conceptually it is the ratio of pressure over coronary flow. In practice this index is calculated using the mean transit time (i.e. a reciprocal of flow; that’s why its units are mmHg x sec)). It is well explained in the main file.

7. PLOS authors have the option to publish the peer review history of their article (what does this mean? ). If published, this will include your full peer review and any attached files.

**Do you want your identity to be public for this peer review?** For information about this choice, including consent withdrawal, please see our Privacy Policy .

Reviewer #1: **Yes: ** Fabrizio D'Ascenzo

Reviewer #2: No

---

## [Author Response · Author response to Decision Letter 1]

21 Feb 2025

We sincerely thank you for the detailed and constructive feedback provided on our manuscript titled “Index microvascular resistance (IMR)-guided management of heart transplantation (IMR-HT study): Study Protocol.” We have carefully addressed all the comments and made the necessary revisions to improve the quality of our work.

We have carefully addressed all the comments and made the necessary revisions to improve the quality of our work. Below, we provide a point-by-point response to the reviewers’ comments, highlighting the changes made in the revised manuscript.

For clarity, the editor and reviewers’ comments are presented in bold, followed by our responses in regular text in the rebuttal letter (attached file). Revised text is highlighted in the manuscript for easy reference.

---

## [Editor Report · Decision Letter 1]

23 Feb 2025

Index microvascular resistance (IMR) in Heart Transplant Patients 

(IMR-HT study): Study Protocol

PONE-D-24-52635R1

Dear Dr. Fuertes Ferre,

We’re pleased to inform you that your manuscript has been judged scientifically suitable for publication and will be formally accepted for publication once it meets all outstanding technical requirements.

Kind regards,

Giuseppe Andò, M.D., Ph.D.

Academic Editor

PLOS ONE
---

## [Editor Report · Acceptance letter]

PONE-D-24-52635R1

PLOS ONE

Dear Dr. Fuertes Ferre,

I'm pleased to inform you that your manuscript has been deemed suitable for publication in PLOS ONE. Congratulations! Your manuscript is now being handed over to our production team.

Kind regards,

on behalf of

Prof. Giuseppe Andò

Academic Editor

PLOS ONE